# Low-Velocity Impact and Post-Impact Residual Flexural Properties of Kevlar/EP Three-Dimensional Angle-Interlock Composites

**DOI:** 10.3390/ma17030681

**Published:** 2024-01-31

**Authors:** Juanjuan Shi, Yanwen Guo, Xiaomei Huang, Hongxia Chen, Haijian Cao

**Affiliations:** School of Textile and Clothing, Nantong University, Nantong 226007, China; 19952613837@163.com (J.S.); 15806290797@163.com (Y.G.); h.xmei@ntu.edu.cn (X.H.); chenhx@ntu.edu.cn (H.C.)

**Keywords:** three-dimensional angle-interlock (3DAI), low-velocity impact property, flexural property after impact (FAI)

## Abstract

In this study, five three-dimensional angle-interlock fabrics with different warp and weft densities were fabricated using 1000D Kevlar filaments. The Kevlar/EP composites were prepared by vacuum-assisted molding techniques. The low-velocity impact property of the composite was tested, focusing on the effects of the warp and weft densities, impact energy, impactor shape, and impactor diameter. The damage area, dent depth, and crack lengths in the warp and weft direction were used to evaluate the impact performance, and the specimens were compared with plain-weave composites with similar areal densities. The dominant failure mode of the conical impactor was fiber fracture, while the dominant failure mode of the hemispherical impactor was fiber–resin debonding. The cylindrical impactor showed only minor resin fragmentation. The residual flexural strength of the composite after impact was tested to provide insights into its mechanical properties. The study findings will provide a theoretical basis for the optimization of the design of impact-resistant structures using such materials and facilitate their engineering applications.

## 1. Introduction

Textile composites have been widely employed in various fields, such as construction engineering and military protection, due to their advantages of a light weight and high strength [1]. The three-dimensional angle-interlock (3DAI) fabric is a three-dimensional (3D) structure formed by the warp yarns passing through the weft yarns in the thickness direction. Compared to traditional two-dimensional (2D) composites, it exhibits excellent interlaminar shear resistance and damage tolerance due to the interlocking effect of the interwoven warp yarns [2,3,4]. Moreover, due to the slip effect of the warp yarns, the 3DAI fabric demonstrates excellent formability, making it frequently utilized in the manufacturing of products with complex curved surfaces, such as helmets [5,6].

During practical applications, composite materials are inevitably subjected to various low-velocity impact loads, which exhibit significant randomness. These impacts are influenced by factors such as different impact energies, impactor shapes/sizes, impact locations, and multiple impacts. Ali Kurşun et al. [7] conducted a study utilizing experimental and finite element simulation techniques to investigate the impact performance of laminates under the influence of four different impactor shapes: hemispherical, conical, cylindrical, and elliptical. The results indicated that the cylindrical impactor exhibited the highest contact force and the shortest contact time, while the conical impactor demonstrated the lowest contact force and the longest contact time. Bulent M I et al. [8] studied the effect of the impactor diameter on the impact performance of glass/epoxy resin composites and found that as the impactor diameter increased, the contact force and penetration threshold also increased. However, within the penetration threshold range, the absorbed energy decreased with the increase in the impactor diameter.

The internal damage of 3DAI composites subjected to out-of-plane impact manifests as matrix cracking and fiber–matrix debonding [9,10]. Such types of damage not only weaken the structure but are also challenging to detect, as they occur at relatively low impact energies and leave no visible traces on the material surface. If undetected, this damage can further propagate under fluctuating stress, leading to a deterioration in the mechanical properties of the composite material and the presence of potential failures [11,12]. Therefore, it is necessary to study the damage tolerance of composite materials.

The parameter used in evaluating and quantifying the damage tolerance of composite materials after impact is the residual strength, such as the compressive strength after impact (CAI) [13,14], tensile strength after impact (TAI) [15,16], and flexural strength after impact (FAI) [17,18]. Traditionally, the CAI has been widely used to evaluate the post-impact performance of composites. Extensive experimental and finite element studies have been conducted, with the research literature primarily focusing on the impact damage mechanisms of composite laminates, the relationship between the residual strength and impact damage, and the influence of the fabric structural parameters on the residual mechanical properties [19,20,21]. However, there is relatively little research on the TAI and FAI [22]. Moreover, many researchers have recently criticized the use of only the CAI to evaluate and quantify the residual bearing capacity, especially for composite components primarily subjected to tensile and flexural loads [22,23,24]. Therefore, it is essential to evaluate the FAI of composites that primarily endure flexural loads [18,24,25].

The test methods for FAI primarily include the three-point bending test and four-point bending test. Among the flexural tests, three-point bending tests are advantageous for local stress localization at the impacted damaged area due to the contact of the loading head, allowing for delamination extension along the entire span length; four-point bending tests suppress delamination propagation once it reaches the loading points [26]. However, some researchers argue that three-point bending has certain limitations since the loading point is located at the area of maximum damage, which may cause additional specimen damage [27]. Both methods have their own advantages and disadvantages in assessing the post-impact residual flexural performance, and the appropriate choice should be made based on the specific circumstances. Tian et al. [17] investigated the influence of the hybridization of carbon fiber and aramid fiber, as well as different laminate configurations, on the low-velocity impact and post-impact flexural performance of composite materials. They analyzed the impact damage mechanism and post-impact flexural damage mechanism. The results demonstrated that when the aramid fiber was interleaved with the carbon fiber and placed on the top and bottom layers, this laminate structure combined the advantages of both fibers, exhibiting excellent impact resistance and residual flexural performance. Wagih et al. [26] tested the impact resistance and FAI of carbon fiber and aramid fiber sandwich hybrid composites, and it was observed that the carbon fibers in the lower layer of the composite laminate remained intact. This could be attributed to the significant deformation and energy absorption of the aramid fibers in the intermediate layer. Sarasini et al. [28] investigated the impact performance and post-impact four-point bending performance of hybrid laminates composed of basalt fiber and aramid fiber. The results indicated that incorporating a suitable hybrid design was beneficial in enhancing the impact resistance and residual flexural strength of the composite laminate. Hart et al. [23] quantitatively evaluated the CAI and FAI of 2D and 3D composites with the same surface density, and it was found that the decrease in post-impact flexural performance was greater than that in the compression performance. This is because the post-impact flexural strength and modulus are more sensitive to the delamination damage caused by impact. Therefore, when assessing post-impact performance, it is crucial to select the appropriate method based on the actual engineering conditions of the product. If the flexural load is the main load-bearing mechanism, the FAI should be employed. On the other hand, if the compression load is the primary load-bearing mechanism, the CAI should be used to evaluate the post-impact performance. This approach enhances the effectiveness and accuracy of post-impact performance assessment [25,29].

In summary, extensive research has been conducted on the low-velocity impact property and residual strength of composites. However, there is relatively limited research on the failure mechanisms of 3DAI composites with different warp and weft densities and various impactor shapes. Furthermore, the investigation of the changes in the bending failure mode before and after impact remains insufficient. Therefore, this study aims to investigate the low-velocity impact property of 3DAI composites. The main focus is to explore the influence of factors such as the warp and weft densities, impact energy, and impactor shape/diameter on the impact performance of these composites. Additionally, a comparative analysis is conducted between the impact failure modes and mechanisms of 3DAI composites and 2D plain composites with similar areal densities. Furthermore, the post-impact residual flexural strength and the changes in flexural failure modes before and after impact are analyzed in detail.

## 2. Materials and Methods

### 2.1. Materials and Specimen Preparation

The 3DAI fabrics with five warp and weft densities and 2D plain fabrics were woven from 1000D Kevlar filaments (Yantai, China). The fabrics were prepared on the SGA598 semi-auto sample loom (Wuxin, China), as shown in Figure 1b, using the structural diagrams of 3DAI shown in Figure 1a. The fabric surface of 30 × 30 is shown in Figure 1c. The structural parameters of the fabrics are shown in Table 1.

The Kevlar/EP composites were prepared by the vacuum-assisted resin infusion process (VARI). First, the epoxy resin (Nantong, China) and curing agent (Changzhou, China) were mixed at the ratio of 4:1 and placed in a vacuum box for 30 min to remove bubbles. Then, the VARI process was used to inject the epoxy resin/hardener mixture into the vacuum bag at an injection pressure until the fabrics were fully impregnated. Finally, the vacuum bag was placed in a drying oven with a temperature of 75 °C for 2 h to obtain composites where the average resin content was approximately 50% and the thickness was about 2.5 mm.

### 2.2. Low-Velocity Impact Test

The low-velocity impact tests were conducted on a self-made drop weight impact testing machine in the laboratory, following the reference standard ASTM D7136 [30]. The total weight of the drop weight was 3 kg, and it was equipped with three different impactor shapes with a diameter of 25 mm, as shown in Figure 2. The impactors were fitted with an accelerometer to record the acceleration during the impact process. After the first impact, the impactor was restrained to avoid multiple impacts on the specimen.

The dimensions of the impact specimens were 150 mm × 100 mm, and each group of samples was subjected to three repeated tests.

### 2.3. Three-Point Bending Test

The extent of damage sustained by composite materials post-impact is intrinsically linked to the density of their warp and weft. By employing bend testing methodologies, the residual strength and stiffness of the three-dimensional angle-interlock composite materials of varying specifications were evaluated. This assessment is crucial in determining the longevity of composite materials and plays a significant role in the structural design optimization. The post-impact residual flexural property was evaluated using the three-point bending test, following the testing standard GB/T 1449-2005 [31]. Due to the anisotropy of the 3DAI fabric in both the warp and weft directions, the impacted region of the composite material was cut into 100 mm × 25 mm bending specimens along the warp and weft directions, respectively. The thickness of the sample was about 2.5 mm. The span-to-thickness ratio of the bending specimens was 16:1, and the loading speed was 2 mm/min. Each group of samples underwent three tests, and the bending loading direction was consistent with the impact direction. The flexural strength and flexural modulus were calculated by Formulas (1) and (2), respectively.
(1)$$\sigma_{f}=\frac{3P\times l}{2b\times h^{2}}$$
(2)$$E_{f}=\frac{\sigma \prime \prime -\sigma ’}{\varepsilon \prime \prime -\varepsilon ’}$$
where $\sigma_{f}$—flexural strength; $E_{f}$—flexural modulus; σ″—flexural stress measured when strain ε″ = 0.0025; σ’—flexural stress measured when strain ε’ = 0.0005.

## 3. Results and Discussion

### 3.1. Low-Velocity Impact Test

#### 3.1.1. The Impact Property of Single-Ply 3DAI Composites

As the material in this study was intended for the fabrication of safety helmet shells, thin-layer composites were utilized. Initially, the impact properties of the composites prepared with a single-ply 3DAI fabric were evaluated, and the post-impact failure morphologies of the 25 mm impactors are depicted in Figure 3.

From Figure 3, it is evident that the single-ply 3DAI composites experienced perforating damage upon impact by a conical impactor. The frontal damage morphology exhibited a more regular circular shape, while the backside displayed a mushroom-like bulge protruding outward. The primary mode of energy absorption was fiber fracture. When impacted by a hemispherical impactor with energy of 3 J, the damage morphology exhibited a distinct cross shape. For impact energies of 6–9 J, all specimens experienced perforating damage, with regular circular damage on the front side corresponding to the diameter of the impactor, and an outward explosion and more extensive destruction on the backside. This indicates that the 3 J impact energy did not reach the material’s penetration threshold, but once the threshold was reached, perforating damage occurred. Consequently, it was observed that single-ply 3DAI composites were prone to perforating failure. Therefore, subsequent experiments were conducted using two-ply 3DAI fabrics for further investigation.

#### 3.1.2. Effect of Fabric Specifications on Impact Property

The composites prepared using the fabric specifications in Table 1 were subjected to impact using a hemispherical impactor with diameter of 25 mm, and the data from the accelerometer were recorded. The impact force was calculated using Formula (3), where m is the mass and a is the acceleration, and the impact force of composites with different warp and weft densities is illustrated in Figure 4.
(3)F=m×a

As shown in Figure 4, the impact force of Kevlar/EP 3DAI composites increases with the increase in the fabric warp and weft density. When the warp density is 24 picks/cm, the weft density increases, and its impact force increases but the increase is small. When the warp density is 30 picks/cm, the impact force increases significantly. This can be attributed to the fact that the increase in the warp and weft densities results in an increase in areal density and a tighter fabric structure, leading to greater resistance to impact force.

Taking the 9 J impact energy as an example, the crack lengths in the warp and weft directions (as shown in Figure 5), the damage area, and the dent depth were tested and analyzed. The summarized results are presented in Table 2.

As can be seen from Table 2, with the increase in the warp and weft density, the crack length, damage area, and dent depth of the Kevlar/EP 3DAI composites decreased in the warp and weft directions. For the 24 × 18 and 24 × 21 specimens, the crack lengths in the warp direction were smaller than those in the weft. The warp and weft crack lengths of the 24 × 24 specimen were close, but the warp crack length was slightly larger than the weft. However, for the 30 × 27 and 30 × 30 specimens, the crack lengths in the warp direction were significantly larger than those in the weft. The main reason was that the weft densities of the 24 × 18 and 24 × 21 specimens were relatively small, making it easier for the weft cracks to propagate. On the other hand, for the 30 × 27 and 30 × 30 specimens, both the warp and weft densities were relatively high, resulting in increased crimp % of the warp yarn (as shown in Figure 6). This increased crimp % led to a higher stress concentration during impact, causing cracks to initiate at the interlocking region of the warp and weft yarns. These cracks then propagated along the warp yarn direction, resulting in a longer length of warp cracks. Stig F and Hallstrom S have also reported the negative effect of the warp crimp percentage on the mechanical properties of 3D composites [32]. These findings contribute to a better understanding of the damage mechanisms and performance of composites under impact loading. The results emphasize the importance of the fabric specifications, particularly the warp and weft density, in determining the extent and direction of crack propagation. This knowledge can aid in the design and optimization of composite structures for enhanced impact resistance.

#### 3.1.3. Effect of Impact Energy on Impact Property

According to formula E = mgh, the impact energy was selected by varying the height from which the impactor was dropped. In this experiment, the impact energies were 3 J, 6 J, and 9 J, respectively. The hemispherical impact force–time curve is shown in Figure 7.

From Figure 7, it can be observed that for thin Kevlar/EP 3DAI composites, both the impact force and contact time increase with the increase in impact energy. This is because, as the impact energy increases, the kinetic energy of the impactor is transferred to the composites, resulting in a higher level of damage and a larger area of damage. As a result, the impact force increases. Additionally, due to the increase in damage depth, the contact time also increases.

The impact resistance characteristics of Kevlar/EP 3DAI composites are as follows: when the hemispherical impactor falls freely from a certain height, the gravitational potential energy of the impactor is transformed into impact kinetic energy, and the kinetic energy of the impactor is transferred to the composite when it comes into contact with the sample. Part of the transmitted energy is absorbed by the composite, resulting in internal damage, and the other part of the energy is stored in the composite in the form of elastic energy, resulting in the phenomenon of the rebound of the impactor. The impact force–time curve can be divided into two stages. In the first stage, the impact force increases with time, and the composite is subjected to the compressive stress of the impactor, resulting in bending deformation. Then, the deformation changes from bending to in-plane stretching, resin matrix deformation, and fiber tension deformation. In the second stage, the impact force slowly decreases and the composite tears along the direction of the fabric, accompanied by a series of in-plane damage; the impact energy is absorbed, and then the impactor bounces back to complete the impact.

#### 3.1.4. Effect of Impactor Shape on Impact Property

The impact property of the 24 × 21 3DAI composite was tested with conical, hemispherical, and cylindrical impactors. The damage morphology and area are shown in Figure 8.

As can be seen from Figure 8, the conical impactor only causes damage in a small area near the tip of the impactor, and the total damage area is small, but it is prone to penetration. The damage shadow area of the hemispherical impactor is large but the dent depth is small. After 3 J and 6 J impacts, there is no visible damage to the cylindrical impactor, and after 9 J energy impact, only a small amount of resin fragmentation caused by the edge of the cylindrical impactor appears on the surface, which is called the “annular damage zone”, and there is no visible damage on the back.

The impact resistance of the composites against the impactor is primarily influenced by the contact area. During low-velocity impact, the elastic deformation of the rigid impactor can be neglected. Therefore, the impact contact area of the impactor penetrating the plate is equivalent to the surface area corresponding to a certain depth of penetration into the plate. The impact contact areas for the three shapes of impactors can be calculated using Formulas (4), (5), and (6), respectively.
(4)$$S_{1}=\frac{\sqrt{3}}{3}\pi d^{2}$$
(5)$$S_{2}=2\pi Rd$$
(6)$$S_{3}=\pi R^{2}$$

In the formula, *S*_1_, *S*_2_, and *S*_3_ are the corresponding impact contact areas of the conical, hemispherical, and cylindrical impactors, respectively; *R* is the radius of the impactor; *d* is the impact penetration depth. Obviously, when the penetration depth is the same, the contact area of the cylindrical impactor is the largest, followed by the hemispherical impactor, and that of the conical impactor is the smallest.

According to Figure 9, it can be observed that the impact forces of different impactor shapes are as follows: cylindrical > hemispherical > conical. This indicates that the cylindrical impactor experiences the highest impact resistance when it is in contact with the composite, due to its larger contact area, resulting in the smallest pressure on the unit contact area. On the other hand, the conical impactor experiences the lowest impact resistance, leading to the highest pressure on the unit contact area. The hemispherical impactor falls between the two in terms of impact resistance and pressure on the unit contact area. These results are similar to those of Kursun A [7].

Taking the 9 J impact as an example, the post-impact damage morphology of different impactor shapes is shown in Figure 10. After the impact of the conical impactor, the front impact point shows a circular depression with fibers protruding outward. The resin fragmentation phenomenon spreads outward along the impact point, and the main failure mode is resin fragmentation and fiber–resin debonding. The backside failure mode is fiber fracture with fiber fibrillation phenomenon. After the impact of the hemispherical impactor, the primary failure mode on the front side is fiber–resin debonding, with cracks propagating along the warp and weft directions of the fabric. The damage on the backside is more pronounced, featuring a significant amount of resin–fiber debonding and a minor amount of fiber fracture. After the impact of the cylindrical impactor, there is a small amount of resin fragmentation at the contact area of the front edge of the impactor, and no visible damage is observed on the backside. From the damage morphology, it can be observed that, except for the case of the cylindrical impactor, the damage on the backside is more severe than on the front side, indicating that the damage pattern of thin-layer composite materials is from bottom to top.

#### 3.1.5. The Effect of Impactor Diameter on Impact Property

For impactors with diameters of 10 mm, 12 mm, and 25 mm, the impact forces of the hemispherical impactor are shown in Figure 11. Taking the 9 J energy impact as an example, the crack length, damage area, and dent depth are shown in Table 3.

The impact forces of the hemispherical impactors with diameters of 10 mm, 12 mm, and 25 mm are shown in Figure 11. As the diameter of the impactor increases, the impact force also increases. However, there is not much difference in the impact forces between the 10 mm and 12 mm diameters. When the diameter of the impactor is 25 mm, the impact force increases significantly, with an increase of 30.2% compared to the 10 mm diameter. With the increasing diameter of the impactor, the crack length in the warp and weft directions increases, as does the damage area. However, the dent depth decreases. There is a negative correlation between the dent depth and the damage area. This is because the larger the diameter of the impactor, the larger the contact area with the material, resulting in lower local stresses near the impact point of the specimen [33].

In conclusion, regardless of the different shapes or diameters of the impactor, the essence lies in the difference in the contact area between the impactor and the composite. The larger the contact area between the impactor and the composite, the earlier the attainment of the peak value and the shorter the duration of the impact process. Conversely, the smaller the contact area between the impactor and the composite, the greater the degree of damage to the composite’s plate after impact, and the more energy is absorbed. This energy is consumed by the bending deformation and internal fiber fracture and fiber–resin debonding of the composite.

#### 3.1.6. The Effect of Fabric Structure on Impact Property

The hemispherical impact of 3DAI and 2D plain composites with similar areal densities was evaluated. From Figure 12a, it can be observed that compared to the Kevlar/EP 2D composites, the Kevlar/EP 3DAI composites exhibited a more significant increase in impact resistance with the increase in impact energy. When the impact energy was 3 J, the 2D structure had an impact load of 1828.5 N, while the 3D structure had a load of 1891.8 N. When the impact energy was 6 J, the 2D structure had a load of 2833.5 N, while the 3D structure had a load of 3052.5 N. When the impact energy was 9 J, the 2D structure had a load of 3647.1 N, while the 3D structure had a load of 4102.8 N.

The impact resistance of 3DAI composites is superior to that of 2D composites, as can be seen from the fracture morphology on the side after impact in Figure 12b. The main failure mode of the 3DAI structure is fiber–resin debonding, while the main failure mode of the 2D composites is delamination. The material first experiences compressive stress when it comes into contact with the impactor. Subsequently, the shock wave travels downwards along the fiber direction through the resin. While the front of the composites experiences compressive stress, the bottom experiences tensile stress due to upward bending, which is transmitted through the fibers and matrix. The compressive and tensile stresses meet in the middle, resulting in shear deformation and a larger damage range in the central layer of the laminate. The 2D composite is prone to delamination due to the lack of through-the-thickness reinforcement. However, the 3DAI structure improves the resistance to delamination and in-plane shear strength of the composite due to the through-the-thickness reinforcement provided by the warp yarn, resulting in only fiber–resin debonding.

### 3.2. Residual Flexural Property after Impact (FAI)

Once deformation damage occurs in the 3DAI structure, the residual strength after impact is crucial for the overall load-bearing capacity and durability of the structure. However, for the thin-layer composites studied in this paper, it is not suitable to assess their residual performance using the CAI. Therefore, the FAI is adopted to study the damage tolerance of the composites.

#### 3.2.1. Flexural Failure Mechanism after Impact

Taking the flexural stress–strain curves and fracture morphology before and after 24 × 21 and 30 × 30 impact as examples, the flexural failure mechanism of 3DAI composites before and after hemispherical impact is analyzed in detail.

As shown in Figure 13, the residual flexural characteristics of Kevlar/EP 3DAI composites can be roughly divided into three stages, although the stress–strain curves exhibit slight differences. In the first stage, the elastic deformation stage, the stress increases linearly with the strain. In the second stage, the damage initiation stage, the stress–strain curve varies differently in the warp and weft directions. In the warp direction, the stress approaches a maximum value and then exhibits a gentle yield plateau, with a significant decrease in slope, indicating a pseudo-plastic fracture characteristic. In contrast, the stress in the weft direction experiences an instant drop with a distinct inflection point, accompanied by the audible sound of matrix cracking and fiber fracture, indicating a brittle fracture characteristic. In the third stage, the crack propagation stage, the stress in the warp direction slowly decreases with increasing strain, while the stress in the weft direction fluctuates and then decreases, ultimately remaining unchanged.

The flexural strength and modulus of the composites decrease after impact; the decrease is greater with increasing impact energy and the remaining flexural strength is smaller. After being impacted with energy levels of 3 J, 6 J, and 9 J, the percentage of remaining flexural strength in the warp direction of the 24 × 21 specimen is 85.6%, 58.4%, and 47.5%, respectively. In the weft direction, the percentages are 84.1%, 51.4%, and 24.4%, respectively. For the 30 × 30 specimen, the percentages in the warp direction are 97.3%, 92.3%, and 83.8%, respectively, while, in the weft direction, they are 99.2%, 84.2%, and 67.2%, respectively. Overall, the remaining flexural strength in the warp direction is greater than that in the weft direction.

Combined with the stress–strain curve (Figure 13) and failure mode (Figure 14), it can be observed that there is no significant flexural failure phenomenon in the warp direction of the 24 × 21 specimen before impact. Instead, there is a slow degradation in stiffness and strength during the bending deformation process. After impact, cracks propagate along both the warp and weft directions, and the lower surface of the crack expansion undergoes tensile stress under a flexural load. Therefore, the flexural failure mode quickly transitions to fiber fracture after impact. In contrast, the main failure mode in the weft direction before impact is fiber breakage. After impact, with an increase in impact energy, the phenomenon of fiber breakage becomes more pronounced, and the cracks caused by the impact lead to fiber fracture occurring at smaller strains, as shown in Figure 13b. Furthermore, the length of the weft crack is longer than the warp crack for the 24 × 21 specimen after impact, resulting in a larger decrease in the weft flexural strength compared to the warp flexural strength.

For the 30 × 30 specimen, the stress–strain curve in the warp direction does not show any significant changes before and after impact. When subjected to a flexural load, the load is transmitted through the warp yarns in the straight section in the length direction and through the bent section in the thickness direction. Due to the greater warp crimp %, the cracks propagate along the warp direction after impact, but they do not have a significant impact on the flexural performance. However, in the weft direction, both before and after impact with 3 J energy, the stress–strain curve initially increases and then suddenly decreases, exhibiting obvious brittle fracture characteristics. The primary failure mode is fiber–matrix debonding. However, as the impact energy increases, a damage propagation stage occurs, and the impact changes the failure mode. Therefore, when subjected to a flexural load, the cracks caused by fiber–matrix debonding have already propagated, and the main failure mode transitions to fiber breakage. Consequently, although the crack length in the warp direction of the 30 × 30 sample is greater than in the weft direction, the decrease in weft flexural strength is still larger than in the warp.

#### 3.2.2. The Residual Flexural Strength after Impact of Different Impactor Shapes

The weft residual flexural strength of 24 × 21 3DAI composites after being impacted by differently shaped impactors is taken as an example, as shown in Figure 15. The results show that the maximum decrease in flexural strength is 81.9% for the conical impactor, 75.6% for the hemispherical impactor, and 4.9% for the cylindrical impactor. The main reason is that after the impact of the conical impactor, fiber fracture occurs, resulting in the lowest residual flexural strength. On the other hand, after the impact of the cylindrical impactor, only a small amount of resin fragmentation is observed, resulting in the highest residual flexural strength. Finally, after the impact of the hemispherical impactor, the specimen has a larger damage area and greater crack propagation, resulting in the residual flexural strength being between those of the other two shapes. These findings provide valuable insights into the effect of different impactor shapes on the mechanical properties of composites. The significant decrease in flexural strength after impact highlights the vulnerability of composite structures to dynamic loading. This knowledge can inform the design and selection of impactors for specific applications, aiming to minimize damage and improve the overall performance of composite structures.

## 4. Conclusions

The low-velocity impact and post-impact residual flexural properties of composites with five different fabric specifications were tested in this study. The mechanisms were investigated by combining curve graphs and the failure morphology, and the conclusions are summarized as follows:(1)The impact force of Kevlar/EP 3DAI composites increases with the increase in the fabric warp and weft densities. The different shapes of impactors have different effects on the damage modes of composite materials. The main failure mode of the conical impactor is fiber fracture, while the main failure mode of the hemispherical impactor is fiber debonding and crack propagation. The main failure mode of the cylindrical impactor is a small amount of resin fragmentation. Additionally, in this study, the damage mode of the composites progresses from bottom to top.(2)After the impact of the hemispherical impactor, cracks propagate along the warp and weft directions. The crack length in the warp direction is smaller than that in the weft direction for the 24 × 18 and 24 × 21 samples. For the 24 × 24 sample, the crack lengths both in the warp and weft directions are close, but the warp crack length is slightly larger than that of the weft. However, when the warp density is 30 picks/cm, the crack length in the warp direction is significantly larger than that in the weft direction.(3)Regardless of the shape or diameter of the impactor, its essence lies in the difference in contact area between the impactor and the composite material. The larger the contact area between the impactor and the composite material, the earlier the peak is reached, and the shorter the duration of the impact process. Conversely, the smaller the contact area, the longer the duration, and the greater the degree of damage to the composite after impact.(4)The flexural strength and modulus of composites decrease after impact, with a greater decrease observed as the impact energy increases, which leads to a decrease in the residual flexural strength. Overall, the residual flexural strength in the warp direction is greater than that in the weft direction. Specifically, the minimum residual flexural strength in the warp direction for the 24 × 21 sample is 47.5%, while, in the weft direction, it is 24.4%. The maximum decrease in flexural strength after impact is 81.9% for the conical impactor, 75.6% for the hemispherical impactor, and 4.9% for the cylindrical impactor.

## Figures and Tables

**Figure 1 materials-17-00681-f001:**
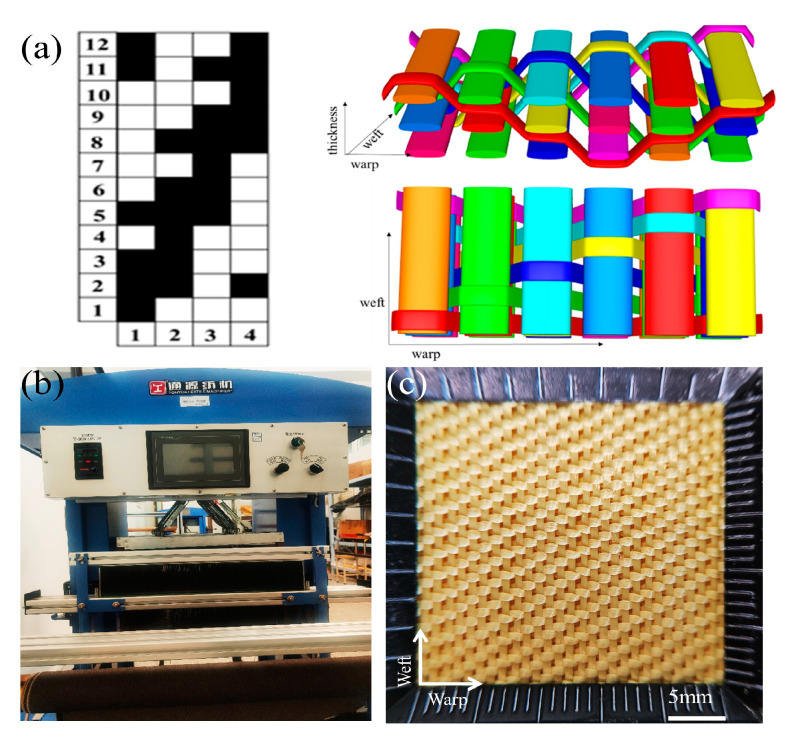
The 3DAI fabric weaving: (**a**) structural diagrams; (**b**) the SGA598 semi-auto sample loom; (**c**) the fabric surface of 30 × 30.

**Figure 2 materials-17-00681-f002:**
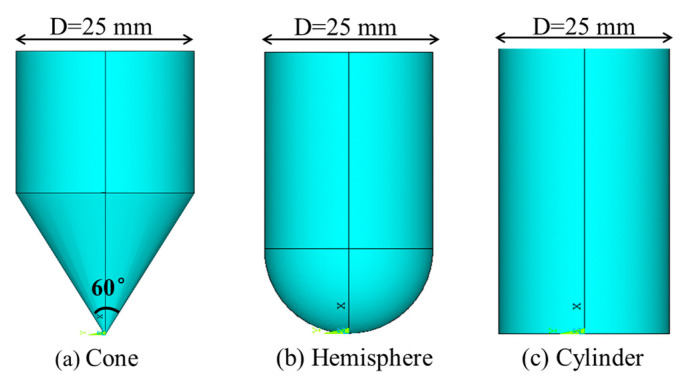
Impactor shape.

**Figure 3 materials-17-00681-f003:**
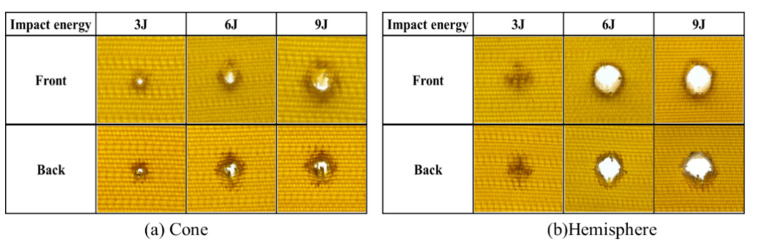
The damage morphology of single-ply 3DAI composites of 24 × 21 after impact.

**Figure 4 materials-17-00681-f004:**
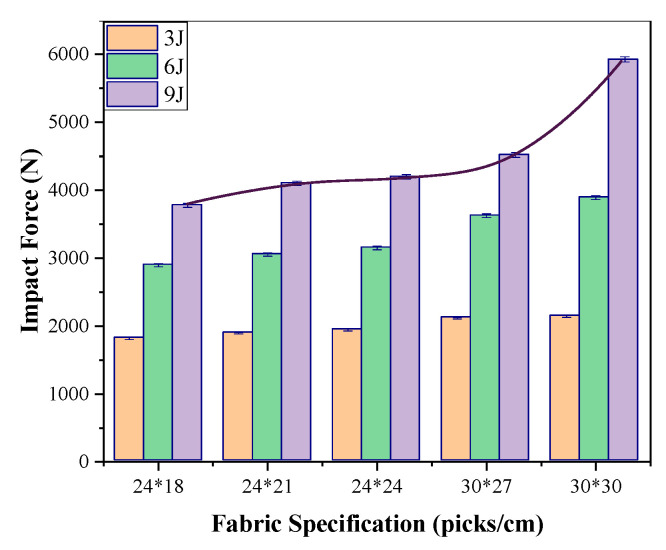
The effect of warp and weft density on impact force.

**Figure 5 materials-17-00681-f005:**
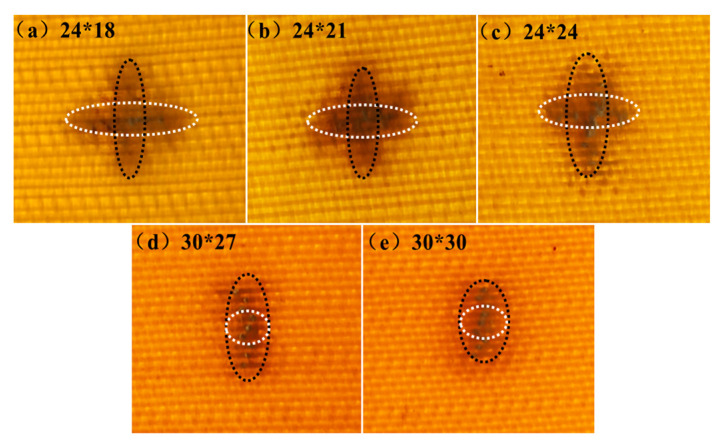
The crack lengths of different fabric specifications (black dotted circle is warp crack length; white dotted circle is weft crack length).

**Figure 6 materials-17-00681-f006:**
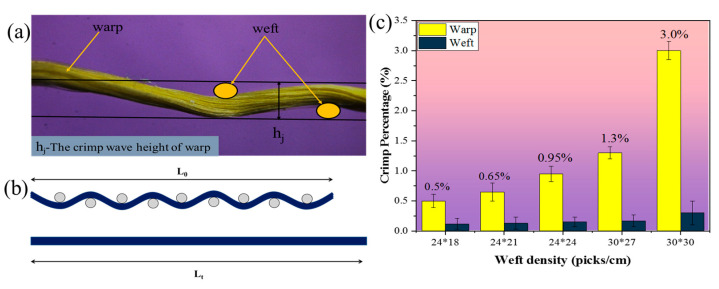
The crimp wave height and crimp percentage: (**a**) the crimp wave height; (**b**) diagram of warp crimp and straightened; (**c**) warp and weft crimp percentage of different fabric.

**Figure 7 materials-17-00681-f007:**
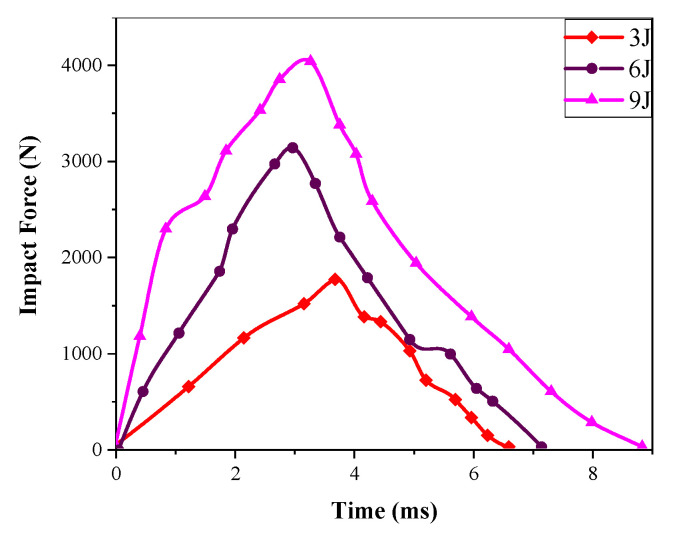
The impact force–time curve of hemisphere impactor.

**Figure 8 materials-17-00681-f008:**
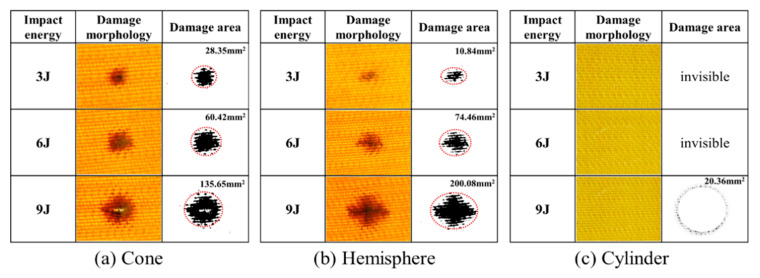
The damage morphology and damage area.

**Figure 9 materials-17-00681-f009:**
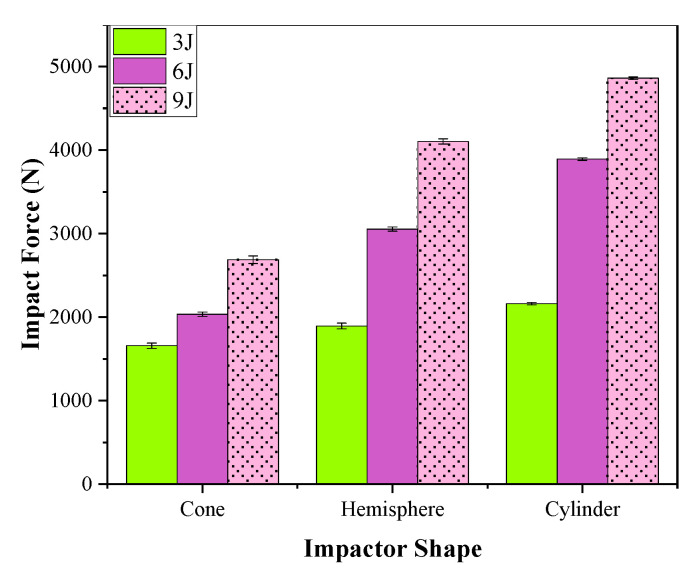
The effect of impactor on the impact force.

**Figure 10 materials-17-00681-f010:**
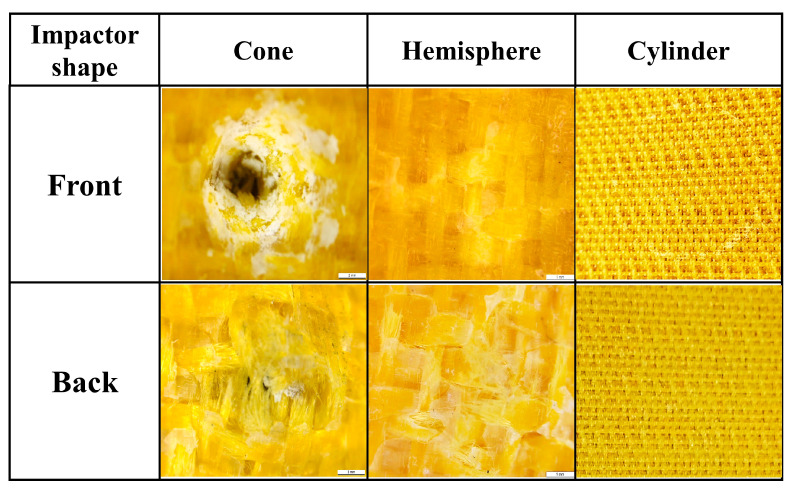
The damage morphology after 9 J energy impact of different impactor shapes.

**Figure 11 materials-17-00681-f011:**
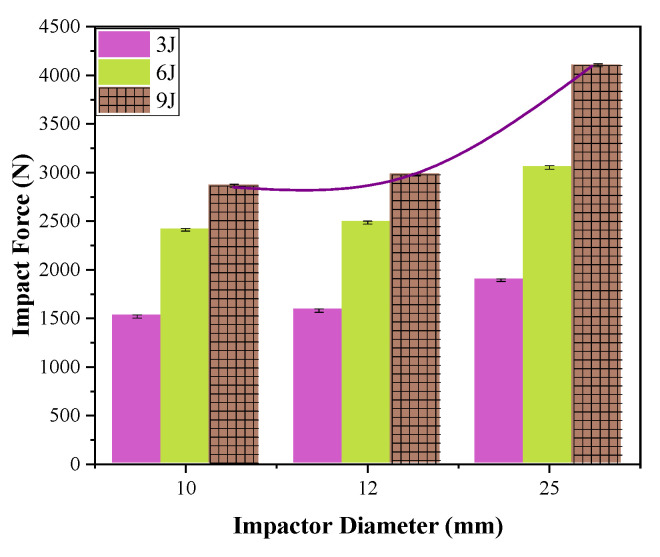
The effect of impactor diameter on impact force.

**Figure 12 materials-17-00681-f012:**
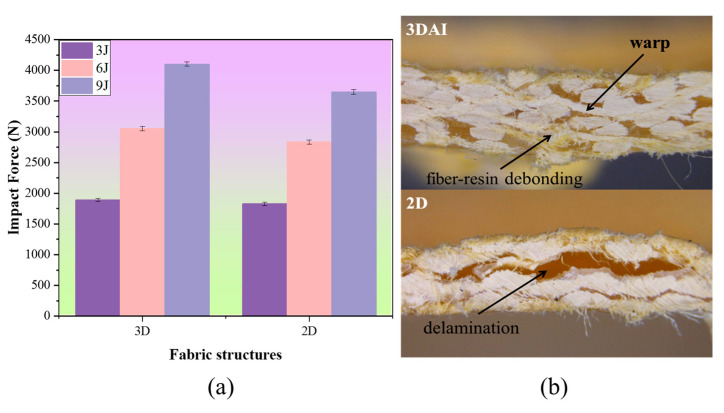
The effect of fabric structures on impact force: (**a**) impact force; (**b**) failure morphology of 3DAI and 2D.

**Figure 13 materials-17-00681-f013:**
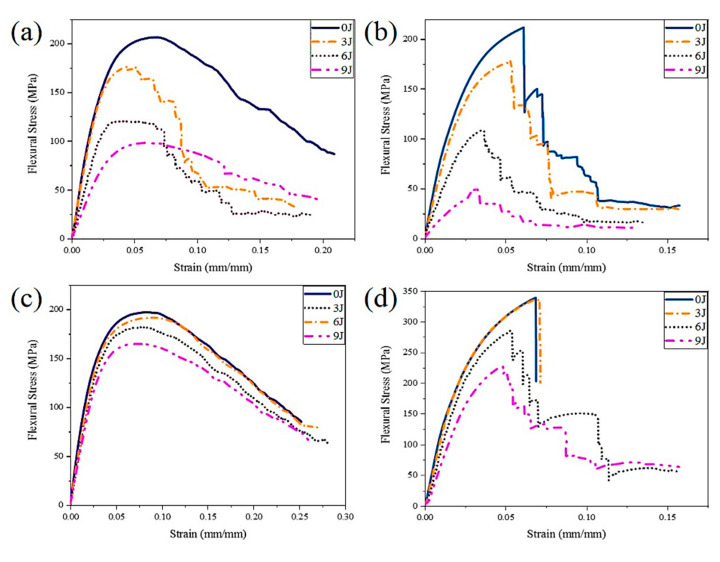
The flexural stress–strain curve before and after hemispherical impact. (**a**) 24 × 21 warp; (**b**) 24 × 21 weft; (**c**) 30 × 30 warp; (**d**) 30 × 30 weft.

**Figure 14 materials-17-00681-f014:**
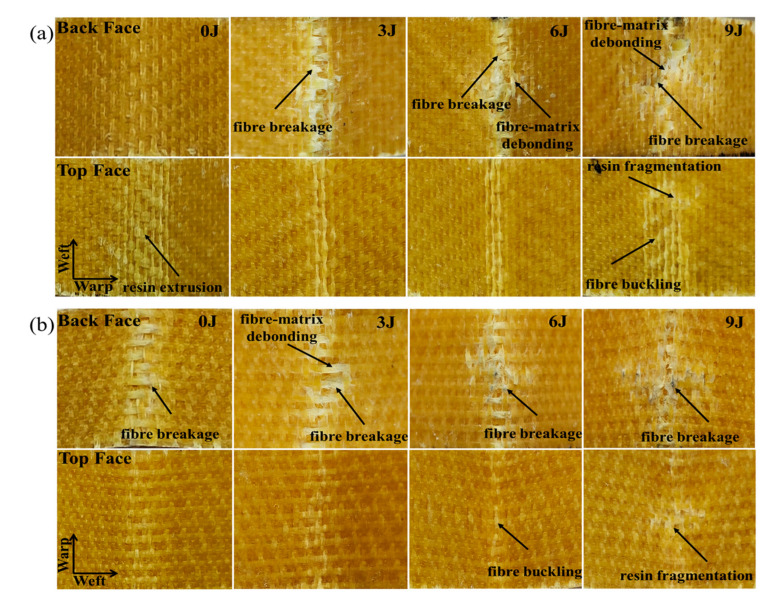
The flexural failure morphology before and after impact of different energy levels: (**a**) 24 × 21 warp; (**b**) 24 × 21 weft; (**c**) 30 × 30 warp; (**d**) 30 × 30 weft.

**Figure 15 materials-17-00681-f015:**
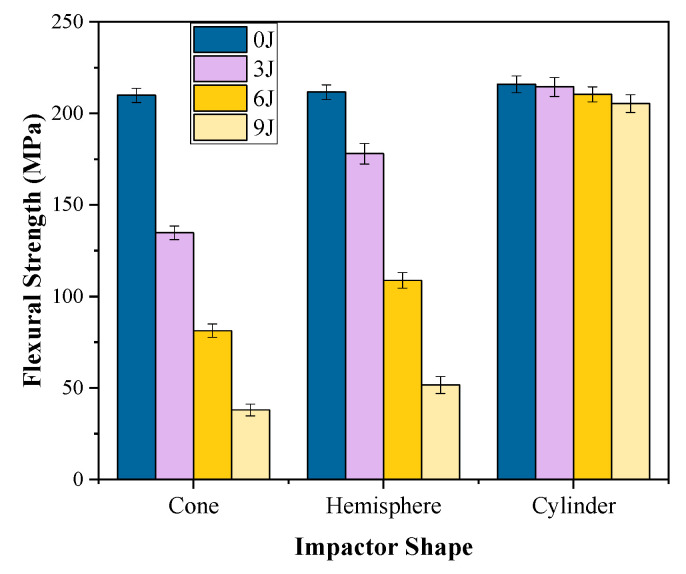
The effect of impactor shape on residual flexural properties after impact.

**Table 1 materials-17-00681-t001:** The Kevlar 3DAI and 2D fabric specifications.

Fabric Specifications	24 × 18	24 × 21	24 × 24	30 × 27	30 × 30	2D
Warp Density (picks/cm)	24	24	24	30	30	9
Weft Density (picks/cm)	24	21	24	27	30	9
No. of Layers (ply)	3	3	3	3	3	3
Fabric Thickness (mm)	1.3	1.3	1.3	1.3	1.3	1.35
Areal Density (g/m^2^)	500	540	580	700	740	560

**Table 2 materials-17-00681-t002:** The effect of fabric specifications on post-impact damage parameters.

Fabric Specification	Crack Length/mm		Damage Area/mm^2^	Dent Depth /mm
	Warp	Weft		
24 × 18	19.3	21.4	300.22	0.62
24 × 21	16.5	18.4	200.08	0.55
24 × 24	13.4	12.6	136.45	0.36
30 × 27	9.2	3.3	24.83	0.20
30 × 30	8.4	2.8	18.48	0.17

**Table 3 materials-17-00681-t003:** The effect of impactor diameter on post-impact damage parameters.

Impactor Diameter	Crack Length/mm		Damage Area/mm^2^	Dent Depth /mm
	Warp	Weft		
10	10.5	12.5	100.25	0.97
12	12.3	15.5	135.68	0.88
25	16.5	18.4	200.08	0.55

## Data Availability

Data are contained within the article.

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
