# Peer review of "Low-Velocity Impact and Post-Impact Residual Flexural Properties of Kevlar/EP Three-Dimensional Angle-Interlock Composites"

_materials, 2024, doi:10.3390/ma17030681_

Round 1

Reviewer 1 Report

Comments and Suggestions for Authors

The paper presents interesting study on the low-velocity impact property of five fabricated three-dimensional composites with different characteristics. After addressing the comments below, it could be considered for publication in Materials:

1. Axis titles on the graphs are not very legible - they should be in a larger font. Similarly, the test on figure 14 is hard to read.

2. The numbering of the observation/results in the text, e.g. (1), (2) is misleading and unusual for scientific articles – it should be removed.

3. Unit of impactor diameter in Table 2 should be included.

Comments on the Quality of English Language

The level of English language is up to the standard of sicentific journals.

Author Response

The paper presents interesting study on the low-velocity impact property of five fabricated three-dimensional composites with different characteristics. After addressing the comments below, it could be considered for publication in Materials:

Response: Thank you very much for finding interest in our research and pointing out the flaws in our paper and giving us an opportunity to revise our manuscript. We have revised the paper according to your comments.

  1. Axis titles on the graphs are not very legible - they should be in a larger font. Similarly, the test on figure 14 is hard to read.

Response: All the axis titles on the graphs have changed to a larger font and Figure 14 has also been modified.

  1. The numbering of the observation/results in the text, e.g. (1), (2) is misleading and unusual for scientific articles – it should be removed.

Response: The text numbers have been removed as requested.

  1. Unit of impactor diameter in Table 2 should be included.

Response: Table 3 is the data tested by impactor with same diameter, and impactor diameter have been given in the first paragraph of 3.1.2. “using a hemispherical impactor with diameter of 25 mm”

Reviewer 2 Report

Comments and Suggestions for Authors

Dear Authors

the publication titled "Low-velocity impact and post-impact residual flexural properties of Kevlar/EP three-dimensional angle-interlock composites" contains a significant number of editing errors related to the font used, its size, the way of placing references, and the presentation of data that are not consistent with the form Materials journal.

Please also refer to any inaccuracies related to the standards used, e.g. Sample dimensions.

Moreover, while the analysis of properties related to impact resistance is absolutely justified, please explain why the mechanical properties of already damaged composites were analyzed?

Please correct the publication.

Best regards

Author Response

Reply to the Review Report (Reviewer 2)

Dear Authors

the publication titled "Low-velocity impact and post-impact residual flexural properties of Kevlar/EP three-dimensional angle-interlock composites" contains a significant number of editing errors related to the font used, its size, the way of placing references, and the presentation of data that are not consistent with the form Materials journal.

Please also refer to any inaccuracies related to the standards used, e.g. Sample dimensions.

Moreover, while the analysis of properties related to impact resistance is absolutely justified, please explain why the mechanical properties of already damaged composites were analyzed?

Please correct the publication.

Best regards

Response: Thank you for your carefully checks and pointing out the flaws in our paper and giving us an opportunity to revise our manuscript. Your comments are very professional and detailed, we have made extensive revisions, the yellow part that has been revised and hope you are satisfied with our revisions. All the comments have been replied point-by-point as follows:

  1. Different font sizes were used.

Response: The font sizes have been changed as requested.

  1. The manner of providing references throughout the document, which is inconsistent with the publication format.

Response: The manner of providing references throughout the document have been changed in accordance with the publication format.

  1. No radius information.

Response: The diameter of the impactors have been shown in Figure 2.

  1. What is the purpose of analyzing the mechanical properties (bending) of samples after the impact properties analysis?

Response: The purpose of analyzing the post-impact flexural strength has been added in the first paragraph in the 2.3. “The extent of damage sustained by composite materials post-impact is intrinsically linked to the density of their warp and weft. By employing bend testing methodologies, the residual strength and stiffness of three-dimensional-angle-interlock composite materials of varying specifications were evaluated. This assessment is crucial for determining the longevity of composite materials and plays a significant role in the enhancement of struc-tural design optimization.”

  1. The unit should be separated from the given value.

Response: The units and given value of the full text have been checked and separated.

  1. What was the thickness of the sample?

Response: The sample thickness varies according to the fabric warp and weft density, but the two composite materials are about 2.5mm, and the data will be processed according to different sizes of each sample in this paper.

  1. The given sample dimensions do not comply with those specified in the GB/T 1449-2005 standard.

Response: The GB/T 1449-2005 standard stipulates that the sample size is 100 mm×15 mm, but the crack length of the sample after impact is greater than 15 mm, so we set the width to 25 mm, and finally calculate the stress to reduce the impact of the size on the bending strength.

  1. The tested material is a composite, in the case of this type of material, the strain rate should be 2 mm/min due to their stiffness, have the authors tested the mechanical properties at other speeds? does the standard used specify the test speed?

Response: Thanks for your careful checks, we are really sorry for our careless mistakes. The test speed was 2 mm/min and has been changed.

  1. Equation (3) description.

Response: Equation (3) has been described in the first paragraph of 3.1.2. “The impact force was calculated using formula (3), where, m is mass; a is acceleration.”

  1. Figure 3 presents other dimensions of impactors

Response: In Figure 3, they were impacted by two impactor with a diameter of 25 mm. Table 3 shows the three diameters of hemispherical impactor.

Reviewer 3 Report

Comments and Suggestions for Authors

The paper is devoted for low velocity impact and post-impact residual flexural properties of Kevlar/EP  three-dimensional angle-interlock composites. The topic is interesting, however the paper contain unexplained places (below) and need major revisions.

From the discussion to Fig. 4 is not clear why at the density 24 picks/cm the increase of impact force is small, why at the warp density 30 picks/cm the corresponding increase is significant.

In the part 3. Results and discussion is a lack of comparison with similar results already obtained in literature.

In the discussion to the Fig. 7, please explain why different color curves are differently asymmetric shaped.

Fig. 13 should be more discussed.

Conclusions should be rewritten in more informative way.

English need minor revisions.

All small errors should be corrected, for example in Abstract line 1 ‘’different different’’. Numbers and measurements units should be written separately, for example ‘’30 picks/cm’’.

Comments on the Quality of English Language

English need minor revisions. Mainly article the use.

Author Response

Reply to the Review Report (Reviewer 3)

The paper is devoted for low velocity impact and post-impact residual flexural properties of Kevlar/EP three-dimensional angle-interlock composites. The topic is interesting, however the paper contain unexplained places (below) and need major revisions.

Response: Thank you very much for finding interest in our research and pointing out the flaws in our paper and giving us an opportunity to revise our manuscript. We have revised the paper according to your comments and we hope the revised paper can be accepted by you.

  1. From the discussion to Fig. 4 is not clear why at the density 24 picks/cm the increase of impact force is small, why at the warp density 30 picks/cm the corresponding increase is significant.

Response: The reason has been given in the second paragraph of 3.1.2, such as the yellow highlight.

  1. In the part 3. Results and discussion is a lack of comparison with similar results already obtained in literature.

Response: Many studies have also reported the negative effect of the warp crimp percentage on the mechanical properties of 3D composite. We have added the same effect from the literature to paragraph 5 of 3.1.2.

  1. In the discussion to the Fig. 7, please explain why different color curves are differently asymmetric shaped.

Response: The reason for the different asymmetric shapes of different color curves is that the wave frequency is different due to the different impact energy.

  1. Fig. 13 should be more discussed.

Response: Figure 13 has been discussed in as much detail as possible.

  1. Conclusions should be rewritten in more informative way.

Response: Conclusions have been rewritten as required.

  1. English need minor revisions.

Response: We have tried our best to refine and improve English in our manuscript according to your advice. We hope English writing in the revised manuscript could be acceptable for you.

  1. All small errors should be corrected, for example in Abstract line 1 “different different’’. Numbers and measurements units should be written separately, for example “30 picks/cm’’.

Response: Thanks for your careful checks, we are really sorry for our careless mistakes. Based on your comments, we have checked and corrected the whole manuscript of small errors in detail. And numbers and measurements units in full paper have been written separately.

Round 2

Reviewer 2 Report

Comments and Suggestions for Authors

Dear Authors

Thank you for submitting the revised version of the manuscript for review and including my comments .

Best regards

TS

Reviewer 3 Report

Comments and Suggestions for Authors

Authors make proper corrections according to reviewer remarks and I suggest

to publish the paper as it is.